# STABLE DISTRIBUTION ALIGNMENT
# USING THE DUAL OF THE ADVERSARIAL DISTANCE

**Ben Usman**
Boston University
usmn@bu.edu

**Kate Saenko**
Boston University
saenko@bu.edu

**Brian Kulis**
Boston University
bkulis@bu.edu

## ABSTRACT

Methods that align distributions by minimizing an adversarial distance between them have recently achieved impressive results. However, these approaches are difficult to optimize with gradient descent and they often do not converge well without careful hyperparameter tuning and proper initialization. We investigate whether turning the adversarial min-max problem into an optimization problem by replacing the maximization part with its dual improves the quality of the resulting alignment and explore its connections to Maximum Mean Discrepancy. Our empirical results suggest that using the dual formulation for the restricted family of linear discriminators results in a more stable convergence to a desirable solution when compared with the performance of a primal min-max GAN-like objective and an MMD objective under the same restrictions. We test our hypothesis on the problem of aligning two synthetic point clouds on a plane and on a real-image domain adaptation problem on digits. In both cases, the dual formulation yields an iterative procedure that gives more stable and monotonic improvement over time.

## 1 INTRODUCTION

Adversarial methods have recently become a popular choice for learning distributions of high-dimensional data. The key idea is to learn a parametric representation of a distribution by aligning it with the empirical distribution of interest according to a distance given by a discriminative model. At the same time, the discriminative model is trained to differentiate between true and artificially obtained samples. Generative Adversarial Networks (GANs) that use neural networks to both discriminate samples and parameterize a learned distribution, have achieved particularly impressive results in many applications such as generative modeling of images (Goodfellow et al., 2014; Denton et al., 2015; Radford et al., 2015), image super-resolution (Ledig et al., 2016), and image-to-image translation (Isola et al., 2016). Adversarial matching of empirical distributions has also shown promise for aligning train and test data distributions n scenarios involving domain shift (Ganin & Lempitsky, 2015; Tzeng et al., 2015; 2017).

However, GANs and related models have proved to be extremely difficult to optimize. It has been widely reported that training GANs is a tricky process that often diverges and requires very careful parameter initialization and tuning. Arjovsky and Bottou (Arjovsky & Bottou, 2017) have recently identified several theoretical problems with loss functions used by GANs, and have analyzed how they contribute to instability and saturation during training.

In this paper, we focus on one of the major barriers to stable optimization of adversarial methods, namely their min-max nature. Adversarial methods seek to match the generated and real distributions by minimizing some notion of statistical distance between the two, which is often defined as a maximal difference between values of certain test (*witness*) functions that could differentiate these distributions. More specifically in the case of GANs, the distance is usually considered to be equal to the likelihood of the best neural network classifier that discriminates between the distributions, assigning "real or generated?" labels to the input points. This way, in order to align these distributions one has to minimize this maximum likelihood w.r.t. the parameters of the learned or aligned distribution.

Unfortunately, solving min-max problems using gradient descent is inherently very difficult. Below we use a simple example to demonstrate that different flavors of gradient descent are very unstable when it comes to solving problems of this kind.

To address this issue, we explore the possibility of replacing the maximization part of the adversarial alignment problem with a dual minimization problem for linear and kernelized linear discriminators. The resulting dual problem turns out to be much easier to solve via gradient descent. Moreover, we make connections between our formulation and existing objectives such as the Maximum Mean Discrepancy (MMD) (Gretton, 2012). We show that it is strongly related to the iteratively reweighted empirical estimator of MMD.

We first evaluate how well our dual method can handle a point alignment problem on a low-dimensional synthetic dataset. Then, we compare its performance with the analogous primal method on a real-image domain adaptation problem using the Street View House Numbers (SVHN) and MNIST domain adaptation dataset pair. Here the goal is to align the feature distributions produced by the network on the two datasets so that a classifier trained to label digits on SVHN does not loose accuracy on MNIST due to the domain shift. In both cases, we show that our proposed dual formulation of the adversarial distance often shows improvement over time, whereas using the primal formulation results in drifting objective values and often does not converge to a solution.

Our contributions can be summarized as follows:
- we explore the dual formulation of the adversarial alignment objective for linear and kernelized linear discriminators and how they relate to the Maximum Mean Discrepancy;
- we demonstrate experimentally on both synthetic and real datasets that the resulting objective leads to more stable convergence and better alignment quality;
- we apply this idea to a domain adaptation scenario and show that the stability of reaching high target classification accuracy is also positively impacted by the dual formulation.

## 2 RELATED WORK

There has been a long line of work on unsupervised generative machine learning, a review of which is beyond the scope of this work. Recently, adversarial methods for learning generative neural networks have achieved popularity due to their ability to effectively model high dimensional data, e.g., generating very realistic looking images. These include the original Generative Adversarial Networks (GANs) (Goodfellow et al., 2014) and follow up work proposing improved formulations such as Wasserstein GANs (Arjovsky et al., 2017) and Conditional GANs (Mirza & Osindero, 2014).

Related ideas have been proposed for unsupervised domain adaptation. Neural domain adaptation methods seek to improve the performance of a classifier network on a target distribution that is different from the original training distribution by introducing an additional objective that minimizes the difference between representations learned for source and target data. Some models align feature representations across domains by minimizing the distance between first or second order feature space statistics (Tzeng et al., 2014; Long & Wang, 2015; Sun & Saenko, 2016). When adversarial objectives are used for domain adaptation, a domain classifier is trained to distinguish between the generated source and target representations, both using the standard minimax objective (Ganin & Lempitsky, 2015), as well as alternative losses (Tzeng et al., 2015; 2017). Instead of aligning distributions in feature space, several models perform generative alignment in pixel-space, e.g., using Coupled GANs (Liu & Tuzel, 2016) or conditional GANs (Bousmalis et al., 2017). These models adapt by "hallucinating" what the raw training images might look like in the target test domain. A related line of work uses adversarial training to translate images in one domain to the "style" of a different domain to create artistic effects (Mathieu et al., 2016). Several recent works perform the alignment in an unsupervised way without aligned image pairs (Zhu et al., 2017; Onsiderations et al., 2017). Other objectives for distribution matching that have been proposed in the literature, including Maximum Mean Discrepancy (Gretton, 2012), f-discrepancy (Nowozin et al., 2016) and others, have also been used for generative modeling (Dziugaite et al., 2015; Li et al., 2015). A single step of our iterative reweighting procedure is similar to instance reweighting methods that were theoretically and empirically shown to improve accuracy in the presence of domain shift. For example, Huang et al. (2007) used sample reweighting that minimized empirical MMD between populations to plug it as instance-weights in weighted classification loss, whereas Gong et al. (2013) did that to choose points for a series of independent auxiliary tasks, so no *iterative* reweighting was performed in both cases.

In general, most statistical distances used for distribution alignment fall into one of two categories: they are either f-divergences (e.g. GAN objective, KL-divergence), or integral probability metrics (IPMs) that are differences in expected values of a test function at samples from different distributions maximized over some function family (e.g. Maximum Mean Discrepancy, Wasserstein distance). In

this work we specifically consider the logistic adversarial objective (f-divergence), show that it is useful to optimize its dual, and present a relation between this adversarial dual objective and MMD, another statistical distance with a test function from reproducing kernel Hilbert space (RKHS).

## 3 A MOTIVATING EXAMPLE

We start with a well-known (Goodfellow, 2016) motivating example of a simple min-max problem to show that, even in this basic case, gradient descent might fail dramatically. Let us consider the simplest min-max problem with a unique solution: finding a saddle point of a hyperbolic surface. Given the function $f(x, y) = xy$, our problem is to solve $\min_x \max_y f(x, y)$, which has a unique solution at $(0, 0)$. Suppose that we want to apply gradient descent to solve this problem. The intuitive analog of the gradient vector that we might consider using in the update rule is defined by the vector field $g(x, y) = (x, -y)$. However, at any given point the vector $g(x, y)$ will be tangent to a closed circular trajectory, thus following this trajectory would never lead to the true solution $(0, 0)$.

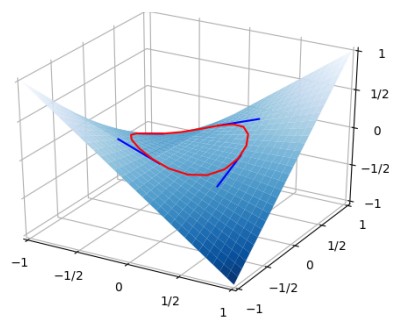

Figure 1: Gradient descent fails to solve the saddle point problem $\min_x \max_y xy$. Red line presents a trajectory of the gradient descent if vector field $g(x, y)$ is used at each iteration. Blue lines are examples of vectors from this vector field.

One can observe the trajectory produced by the update rule $x_{t+1} = x_t + \alpha g(x, y)$ applied to the problem above in Fig. 1. Neither block coordinate descent nor various learning rate schedules can significantly improve the performance of the gradient descent on this problem.

We want to mention that while there is a huge body of work on using alternative descent schemes for convex-concave saddle point optimization, including, but not limited to different variants of mirror descent, such as Nesterov's dual averaging (Nesterov, 2009) and mirror prox (Nemirovski, 2004), authors are not aware of any successful attempts to use it in the context of adversarial distribution alignment. Some techniques developed for solving continuous games such as fictitious play were successfully adopted by Salimans et al. (2016).

## 4 APPROACH

We first propose a new formulation of the adversarial objective for distribution alignment problems. Then we apply this approach to the domain adaptation scenario in Section 5.

Suppose that we are given a finite set of points $A$ sampled from the distribution $p$, and a finite set of points $B$ sampled from the distribution $q$, and our goal is to match $q$ with $p$ by aligning $B$ with $A$. More specifically, we aim to learn a matching function $M_\theta(B)$ that maps $B$ to be as close as possible to $A$ by minimizing some empirical estimate of a statistical distance $d(\cdot, \cdot)$ between them where $\theta$ are parameters of the matching function: $\theta^* = \arg\min_\theta d(A, M_\theta(B))$.

Let us denote $B'_\theta = M_\theta(B)$ or just $B'$ in contexts where dependence on $\theta$ is not important. The regular adversarial approach obtains the distance function by finding the best classifier $D_w(x)$ with parameters $w$ that discriminates points $x \in A$ from points $x \in B'$ and considers the distance between $A$ and $B'$ to be equal to the likelihood of this classifier. A higher likelihood of separating $A$ from $B'$ means that $A$ is far from $B'$. This can be any form of hypothesis in general, and is often chosen to be a linear classifier (Tzeng et al., 2014) or a multi-layer neural network (Goodfellow et al., 2014). In this work, we use the class of linear classifiers, specifically, logistic regression in its primal and dual formulations. The solution can also be kernelized to obtain nonlinear discriminators.

We will define the distance between distributions to be equal to the maximum likelihood of the logistic classifier parametrized by $w$:

$$d(A, B') = \max_w \sum_{x_i \in A} \log(\sigma(w^T x_i)) + \sum_{x_j \in B'} \log(1 - \sigma(w^T x_j)) - \frac{\lambda}{2} w^T w$$

We can equivalently re-write this expression as:

$$C_\theta = \{(x_i, y_i) \; : \; x_i \in A \cup B'_\theta \; , \; y_i = 1 \text{ if } x_i \in A \text{ else } -1\}$$

$$\min_\theta d(A, B'_\theta) = \min_\theta \max_{w,b} \sum_{x_i, y_i \in C_\theta} \log(\sigma(y_i(w^T x_i + b))) - \frac{\lambda}{2} w^T w \tag{1}$$

The duality derivation (Jaakkola & Haussler, 1999; Minka, 2003) follows from the fact that the log-sigmoid has a sharp upper-bound

$$\log(\sigma(u)) \leq \alpha^T u + H(\alpha) \; , \; \alpha_i \in [0, 1]$$

$$H(\alpha) = \alpha^T \log \alpha + (1 - \alpha)^T \log(1 - \alpha)$$

thus we can upper-bound the distance as

$$d(A, B'_\theta) = \min_{0 \leq \alpha \leq 1} \max_{w,b} \sum_{x_i, y_i \in C_\theta} \alpha_i y_i (w^T x_i + b) + H(\alpha) - \frac{\lambda}{2} w^T w$$

where the dual variable $\alpha_i$ corresponds to the weight of the point. Higher weight means that the point is contributing more to the decision hyperplane. Optimal value of alpha attains this upper bound.

The $w$ that maximizes the inner expression can be computed in a closed form, $w^* = \frac{1}{\lambda}(\sum_j x_j y_j \alpha_j)$. Optimally of bias requires $\sum_i \alpha_i y_i = 0$. By substituting $w^*$ we obtain a minimization problem:

$$d(A, B'_\theta) = \min_{0 \leq \alpha_i \leq 1} \frac{1}{2\lambda} \sum_{ij} \alpha_i \alpha_j (y_i x_i)^T (y_j x_j) + H(\alpha) = \min_{0 \leq \alpha_i \leq 1} \frac{1}{2\lambda} \alpha^T Q \alpha + H(\alpha) = \tag{2}$$

$$= \min_{0 \leq \alpha_i \leq 1/\lambda} \frac{1}{2} \alpha_A^T Q_{AA} \alpha_A + \frac{1}{2} \alpha_B^T Q_{BB} \alpha_B - \alpha_A^T Q_{AB} \alpha_B + H(\alpha_A) + H(\alpha_B) \tag{3}$$

$$\text{s.t. } ||\alpha_A||_1 = ||\alpha_B||_1$$

The Eq. (3) is obtained by splitting the summation into blocks that include samples only from $A$, only from $B$, and from both $A$ and $B$. For example, matrix $Q_{AB}$ consists of pairwise similarities between points from $A$ and $B$, and is equal to the dot product between corresponding data points in the linear case. The factor of two in front of the cross term comes from the fact that off-diagonal blocks in the quadratic form are equal. The constraint on alpha sums comes from splitting optimality conditions on the bias into two term. We will denote the resulting objective as $d_D(\alpha, A, B)$.

The above expression gives us a tight upper bound on the likelihood of the discriminator. Thus, by minimizing this upper bound, we can minimize the likelihood itself, as in the original loss, and therefore minimize the distance between the distributions:

$$\theta^*, \alpha^* = \operatorname*{argmin}_{\theta, \alpha \in \mathcal{A}} d_D(\alpha, A, M_\theta(B)) \tag{4}$$

Note that the overall problem has changed from an unconstrained saddle point problem to a smooth constrained minimization problem, which ultimately converges when gradient descent has a properly chosen learning rate, whereas the descent iterations for the saddle point problem are not guaranteed to converge at all.

The resulting smooth optimization problem consists of minimization over $\alpha$ to improve classification scores and over $\theta$ to move points towards the decision boundary. Next section provides more intuition behind the resulting iterative procedure.

## 4.1 RELATIONSHIP TO MMD

In this section, we show that our dual formulation of the adversarial objective has an interesting relationship to another popular alignment objective. The integral probability metric between distributions $p$ and $q$ with a given function family $\mathcal{H}$ is defined as

$$d(p, q) = \sup_{f \in \mathcal{H}} \left| \mathbb{E}_{x \sim p} f(x) - \mathbb{E}_{x \sim q} f(x) \right|.$$

It was shown to have a closed form solution and a corresponding closed form empirical estimator if $\mathcal{H}$ is a unit ball in reproducing kernel Hilbert space with the reproducing kernel $k(x, y)$ and is commonly referred to as Maximum Mean Discrepancy (Gretton, 2012):

$$d(p, q) = \frac{1}{2}\mathbb{E}_{p \times p}k(x, x') + \frac{1}{2}\mathbb{E}_{q \times q}k(y, y') - \mathbb{E}_{p \times q}k(x, y)$$

$$d(A, B) = \frac{1}{2|A|}\sum_{i,j \in A} k(x, x') + \frac{1}{2|B|}\sum_{i,j \in B} k(y, y') - \frac{1}{|A||B|}\sum_{A \times B} k(x, y).$$

From the definition, it is essentially the distance between means of vectors from $p$ and $q$ embedded into the corresponding RKHS. The resulting empirical estimator combines average inner and outer similarities between samples from the two distributions and goes to zero as the number of samples increases if $p = q$.

Note that if sample weights in Eq. (3) are constant and equal across all samples, so $\alpha_i = c$, then the dual distance introduced above becomes exactly an empirical estimate of the MMD plus the constant from the entropic regularizer. Thus, the adversarial logistic distance introduced in Eq. (3) can be viewed as an *iteratively reweighted* empirical estimator of the MMD distance. Intuitively, what this means is that the optimization procedure consists of two alternating minimization steps: (1) find the best sample weights assignment by changing $\alpha$ so that the regularized weighted MMD is minimized, and then (2) use a fixed $\alpha$ to minimize the resulting *weighted* MMD distance by changing the matching function $M_\theta$. This makes the resulting procedure similar to the Iteratively Reweighted Least Squares Algorithm (Green, 1984) for logistic regression. An interesting observation here is that it turns out that high weights in this iterative procedure are given to the most mutually close subsets of $A$ and $B'$, where closeness is measured in terms of Maximum Mean Discrepancy. These happen to be exactly support vectors of the corresponding optimal domain classifier. Therefore, the procedure described above essentially brings sets of the support vectors of the optimal domain classifier from different domains closer together.

We note that the computational complexity of a single gradient step of the proposed method grows quadratically with the size of the dataset because of the kernelization step. However, our batched GPU implementation of the method performed on par with MMD and outperformed primal methods, probably because inference in modern neural networks requires so many dot products that a batch size $\times$ batch size multiplication is negligibly cheap compared to the rest of the network with modern highly parallel computing architectures.

## 5 APPLICATION TO DISTRIBUTION ALIGNMENT IN DOMAIN ADAPTATION

We now show how the above formulation can be applied to distribution alignment for the specific problem of unsupervised domain adaptation. In this scenario, we train our classifier in a supervised fashion on some domain A and have to update it to perform well on a different domain B without using any labeled samples from the latter. Common examples include adapting to a camera with different image quality or to different weather conditions.

More rigorously, we assume that there exist two distinct distributions on $\mathcal{X} \times \mathcal{Y}$: a source distribution $P_S(X, Y)$ and a target distribution $P_T(X, Y)$. We assume that we observe a finite number of labeled samples from the source distribution $D_S \subset [\mathcal{X} \times \mathcal{Y}]^n \sim P_S(X, Y)$ and a finite number of unlabeled samples from the target distribution $D_T \subset [\mathcal{X}]^m \sim P_T(X)$. Our goal then is to find a labeling function $f : \mathcal{X} \to \mathcal{Y}$ from a hypothesis space $\mathcal{F}$ that minimizes target risk $\mathcal{R}_T$, even though we only have labels for samples from source.

$$\mathcal{R}_T(f) = \mathbb{E}_{(x,y) \sim P_T}L(f(x), y) \leq \mathcal{R}_S(f) + d(P_T, P_S) + C(V, n)$$

Ben-David et al. (2007) showed that, under mild restrictions on probability distributions, the target risk is upper-bounded by the sum of three terms: (1) the source risk, (2) the complexity term involving the dataset size and the VC-dimensionality of $\mathcal{F}$, and (3) the *discrepancy* between source and target distributions. Thus, in order to make the target risk closer to the source risk, we need to minimize the discrepancy between distributions. They define discrepancy as a supremum of differences in measures across all events in a given $\sigma$-algebra: $d(p, q) = \sup_{A \in \Sigma} |p(A) - q(A)|$. Estimation of the indicated expression is hard in practice, therefore it is usually replaced with more computationally feasible statistical distances. The total variation between two distributions and the Kolmogorov-Smirnov test

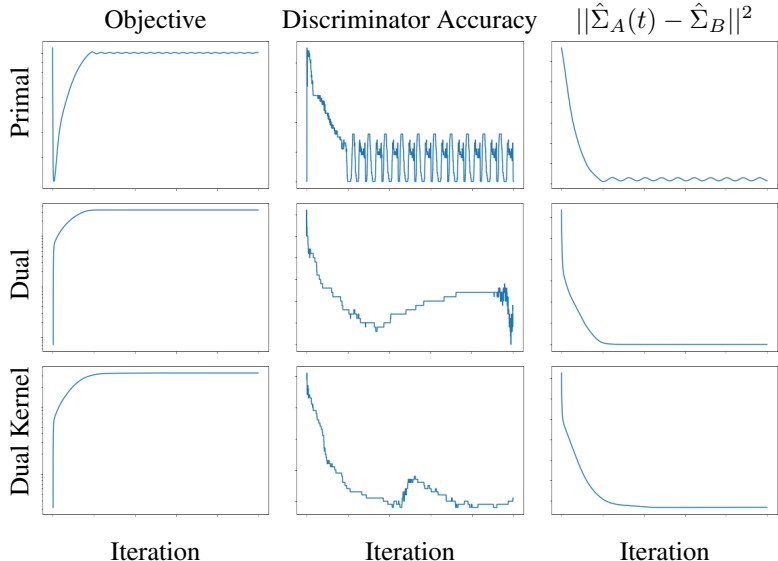

Figure 2: Convergence analysis for the synthetic point alignment problem. **Top row:** The procedure of adversarial alignment in the primal discriminator weight space for points from $A$ and $B$ lying in a two-dimensional plane does not converge to a single solution and oscillates over time. **Middle row:** In contrast, the proposed dual minimization adversarial objective steadily converges to an optimum. **Bottom row:** We observe similar behavior for the kernelized version. Even though the accuracy of the learned discriminators (column two) drifts over time in all cases, the distance between the covariance matrices (column three) and means (not shown in the figure) decreased in all cases; however, in both cases marked as "dual" iterative procedure converged to a single stable solution that indeed corresponds to the desired point cloud alignment. See Figure 3 for an illustration. Note that absolute values of objectives are not comparable, and therefore not shown on plots above.

are closely related to the discrepancy definition above and are also often considered to be too strong to be useful, especially in higher dimensions.

Our approach can be applied directly to this scenario if the discrepancy is replaced with an adversarial objective that uses a logistic regression domain classifier. In Section 6, we consider an instance of this problem where the main task is classification and the hypothesis space corresponds to multi-layer neural networks. We compare the standard min-max formulation of the adversarial objective in Eq. (1) with our min-min formulation in Eq. (3), and report the accuracy of the resulting classifier on the target domain.

## 6 EXPERIMENTS

**Synthetic Distribution Matching**

We first test the performance of our proposed approach on a synthetic point cloud matching problem. The data consists of two clouds of points on a two-dimensional plane and the goal is to match points from one cloud with points from the other. There are no restrictions on the transformation of the target point cloud, so $M_\theta$ includes all possible transformations, and is therefore parameterized by the point coordinates themselves, so the coordinates themselves were updated on each gradient step. We minimized the logistic adversarial distance in primal space by solving the corresponding min-max problem in Eq (1) and compared this to maximizing the proposed negative adversarial distance given by the dual of the logistic classifier in Eq (3) and the corresponding kernelized logistic classifier with a Gaussian kernel.

As expected, the optimization of distances given by the dual versions of domain classifiers (linear and kernelized) worked considerably better than the same distance given by a linear classifier in the primal form. More specifically, the results in the primal case were very sensitive to the choice of learning rate. In general, the resulting decent iterations for the saddle point problem did not converge to a single solution, whereas both dual versions successfully converged to solutions that matched the two clouds of points both visually and in terms of means and covariances, as presented in Fig. 2.

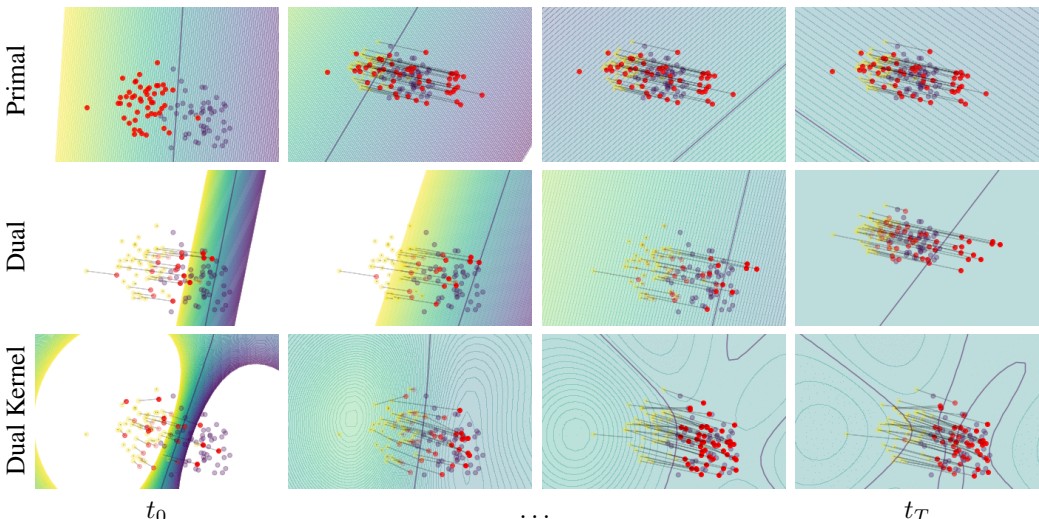

Figure 3: (Best viewed in color) When trained on a point cloud matching task, the primal approach leads to an unstable solution that makes the decision boundary *spin* around data points when they are almost aligned, whereas both the linear and kernel dual approaches lead to stable solutions that gradually assign 0.5 probability of belonging to either $A$ or $B$ to all points, which is exactly the desired behaviour. Yellow and blue points are the original point clouds, red points correspond to positions of yellow points after transformation $M_\theta$.

We suggest one more intuitive explanation of why the dual procedure might work better, in addition to the fact that optimization problems are just inherently easier than saddle point problems. The descision boundary of the classifier in the dual space is defined implicitly through a weighted average of observed data points, so when these data points move, the decision boundary moves with them. If points move too rapidly and the discriminator explicitly parametrizes the decision boundary, weights of the discriminator may change drastically to keep up with moved points, leading to the overall instability of the training procedure. In support of this hypothesis, we observed interesting patterns in the behavior of the linear primal discriminator: when point clouds become sufficiently aligned, the decision boundary starts "spinning" around these clouds, slightly pushing them in corresponding directions. In contrast, both dual classifiers end up gradually converging to solutions that assigned each point with 0.5 probability of corresponding to either of two domains.

**Real-Image Unsupervised Domain Adaptation**

We also evaluated the performance of the proposed dual objective on a visual domain adaptation task. We performed a series of experiments on an SVHN-MNIST digit classification dataset pair in an unsupervised domain adaptation setup: the task is to use a classifier trained on SVHN and unlabeled samples from MNIST to improve test accuracy on the latter.

Following Tzeng et al. (2017) we used standard LeNet as a base model and outputs of the last layer before the softmax as feature representations. We trained the source network to perform well on source dataset and the discriminator to distinguish features computed by the source and target networks. After training the source network on the source domain, we initialized the target network with source weights and optimized it to make the distributions of source and target feature representations less distinguishable from the discriminator perspective. More technical details are given in the Supplementary Section 9.2.

We tested several primal objectives based on Adversarial Discriminative Domain Adaptation (ADDA) (Tzeng et al., 2017), Improved Wasserstein GAN-based objective with a unit-norm gradient regularizer (Salimans et al., 2016), and MMD (Long & Wang, 2015), and compared them to our Dual objective. To eliminate the influence of a particular discriminator and examine the stability of the *objective structure*, we restricted the discriminator hypothesis space $\mathcal{H}$ to linear classifiers, because primal objectives (ADDA, Improved WGAN) cannot be kernelized and MMD does not support multilayer discriminators. This restriction limits the power of the resulting discriminator, thus leading to scores lower than reported state of the art (usually with carefully chosen hyperparameters), but we are more interested in trends in the behaviour of these objectives rather than in absolute reached values.

For each model, we varied learning rates and regularization parameters and ran each experiment for 50 epochs to examine the behavior of these models in the long run. More details on choice of hyperparameters is given in Section 9.1). In unsupervised domain adaptation, we do not have access to target labels and thus cannot perform validation of stopping criteria. In fact, if labeled target data were available then it could be used for fine-tuning the source model, rather than just doing unsupervised learning. Therefore we evaluate the behavior of the models over multiple training epochs to see which would be more stable in the face of uncertain stopping criteria in practical domain adaptation scenarios.

Figure 4 shows the digit classification accuracies obtained by the four models on the target MNIST dataset. The top row presents the distribution of accuracies at different epochs and the bottom row shows the evolution of individual runs. From these results we see that on average descent iterations with our Dual objective converged to satisfactory solutions under a considerably higher number of learning rate and hyperparameter combinations compared to other methods. Our model often stayed at peak performance, whereas all other methods most often slowly deviated from it. The amount of instability demonstrates how important it is to choose exactly the right hyperparameters and stopping criteria for these models. In contrast, our Dual objective (third column) clearly performs well under the majority of the learning rates. WGAN often performs better than MMD and ADDA, but experiences significant oscillations. We tried using different validation heuristics, such as considering only runs that resulted in a significant drop in distance, but this did not significantly change these trends (Section 9.3).

We conclude that our Dual method leads to more stable optimization without the need for choosing an optimal stopping criterion and learning rates by cross-validation on test data.

## 7 DISCUSSION AND FUTURE WORK

While exact dual exists only in the logistic discriminator case, when one can use the duality and solve the inner problem in closed form, we want to stress that our paper presents a more general framework for alignment that can be extended to other classes of functions. More specifically, one can rewrite the quadratic form in kernel logistic regression (3) as a Frobenius inner product of a kernel matrix $Q$ with a symmetric rank 1 alignment matrix $S$ (outer product of alpha with itself).

$$d(A, B) = \min_{0 \leq \alpha_i \leq 1} \frac{1}{2\lambda} \langle \alpha \alpha^T, Q \rangle_F + H(\alpha) = \min_{S_\alpha} \langle S_\alpha, Q \rangle_F + H(S_\alpha)$$

The kernel matrix specifies distances between points and $S$ chooses pairs that minimize the total distance. This way the problem reduces to "maximizing the maximum agreement between the alignment and similarity matrices" that in turn might be seen as replacing "adversity" in the original problem with "cooperation" in the dual maximization problem. In our paper, S is a rank 1 matrix, but we could choose different alignment matrix parameterizations and a corresponding regularizer that would correspond to having a neural network discriminator in the adversarial problem or a Wasserstein distance in Earth Mover's Distance form. The resulting problem is not dual to minimization of any existing adversarial distances, but exploits same underlying principle of "iteratively-reweighted alignment matrix fitting" discussed in this paper.

We assert that in order to understand the basic properties of the resulting formulation, an in-depth discussion of the well-studied logistic case is no less important than the discussion involving complicated deep models, which deserves a paper of its own. This paper proposes a more stable "cooperative" problem reformulation rather than a new adversarial objective as many recent papers do.

## 8 CONCLUSION

We presented an adversarial objective that does not lead to a min-max problem. We proposed using the dual of the discriminator objective to improve the stability of distribution alignment, showed its connection to MMD, and presented quantitative results for alignment of toy datasets and unsupervised domain adaptation results on real-image classification datasets. Our results suggest that the proposed dual optimization objective is indeed better suited for learning with gradient descent than the saddle point objective that naturally arises from the original primal formulation of adversarial alignment. Further attempts to use duality to reformulate other notions of statistical distances in adversarial settings as computationally feasible minimization problems may be promising.

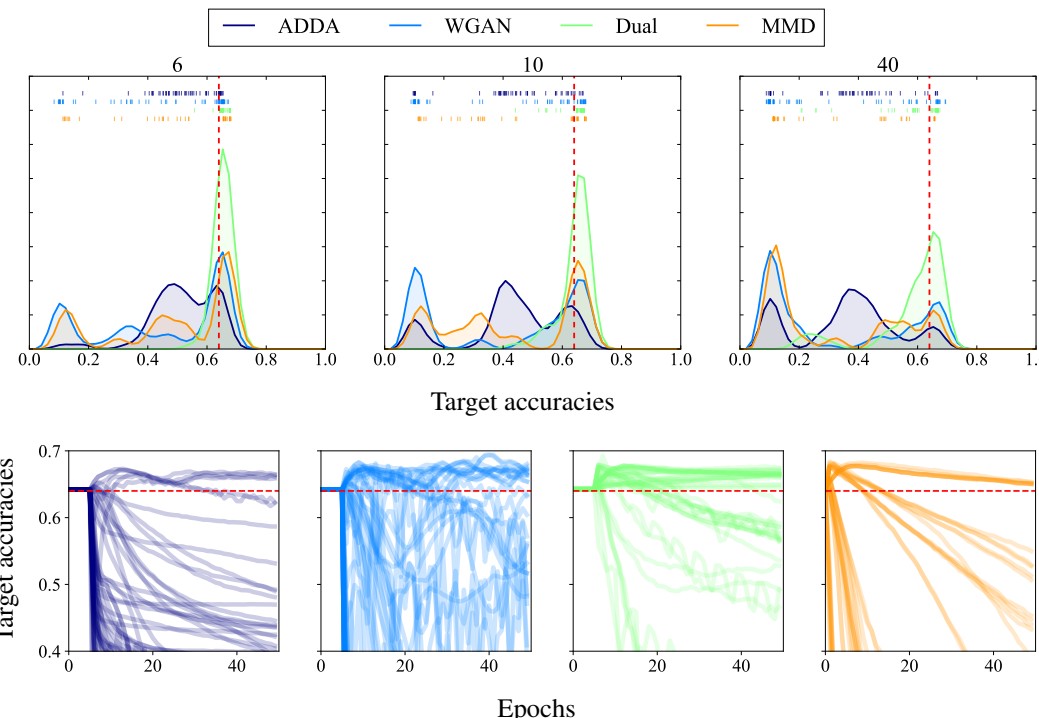

Figure 4: (Best viewed in color) **Top row:** Distribution of target test accuracies at different epochs with different objectives during SVHN-MNIST domain adaptation. The red dashed line represents source accuracy, therefore, a larger accuracy distribution mass to the right of (above) the red line is better. These results suggest that our Dual objective leads to very minimal divergence from the optimal solution under the majority of learning rates and hyperparameter combinations. The other methods have lower solution stability, in descending order: Improved WGAN, MMD, ADDA. **Bottom row:** Evolution of target test accuracy over epochs. Our Dual objective (third column) clearly performs well under the majority of the learning rates. WGAN often performs better than MMD and ADDA, but experiences significant oscillations. Different validation heuristics, such as considering only runs that resulted in a significant drop in distance, did not significantly change these trends (Section 9.3). The proportion of runs that outperformed the source baseline after 40 epochs were: 52.3% for Dual, 21.5% for WGAN, 17.1% for MMD and 6.9% for ADDA.

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

## 9 SUPPLEMENTARY

### 9.1 HYPERPARAMETER SPACE

The space of hyperparameters (mostly, learning rates for different parts of the network) we explored in those runs was built as follows:

1) on SVHN-MNIST we found reasonably working vectors of hyper-parameters for each method essentially by cross-validation on test

2) then we combined all these vectors into a set $H$ and then inflated $H$ by adding many combinations of parts from these vectors and other similar and in-between vectors; for example, if for two models two optimal sets of parameters included (1e-6, 1e-3) for the first model and (1-4, 1e-1) for the second one, we would inflate H by adding all nine combinations of ([1e-6, 1e-5 1e-4], [1e-3, 1e-2, 1e-1]) and varying model-specific parameters such as regularization parameters

3) then we conducted experiments with combinations of pairs [vector from $H$, model]

### 9.2 IMPLEMENTATION DETAILS

During domain adaptation experiments, discriminators of all models that have a trainable discriminator (ADDA, WGAN, Dual) were trained for five epochs prior to the beginning of actual adaptation, which improved performance for all models. Kernel density estimates on Figure 4 were obtained with a fixed 0.03 bandwidth and a linear boundary correction method. All algorithms took greyscale images resized to $28 \times 28$ as input. Restrictions on $\alpha$ were imposed by adding an error term $+\lambda_1 |\sum \alpha_A - \sum \alpha_B|$ and positivity by $-\lambda_2 |\min(0, \alpha)|$. Hyperparameters for the Dual method can be set so that restrictions above hold by the end of adaptation. To implement the dual algorithm in mini-batch fashion we extracted slices of $\alpha_*$ that correspond to points processed in this batch on each iteration. Search space was built as explained above, learning rates and weight decay for updating target network and discriminator were as follows:

- for target network update we used Adam with lr in [1e-6, 1e-5, 1e-4] and beta in [0.5, 0.9]

- for discriminator updates we used SGD with lr in [1e-6, 1e-5, 1e-7] or Adam with learning rates in [2e-4, 1e-6, 1e-5] and beta in [0.5, 0.9]

- discriminator weight decay parameter was chosen from [2.5e-05, 0]

resulting into 90 possible combinations + model specific parameters.

### 9.3 VALIDATION HEURISTICS

Considering runs such that the following holds for different values of $\lambda$ did not change results significantly.

$$\frac{d(P, Q^{(0)}) - d(P, Q^{(T)})}{\max_t d(P, Q^{(t)}) - \min_t d(P, Q^{(t)})} > \lambda.$$

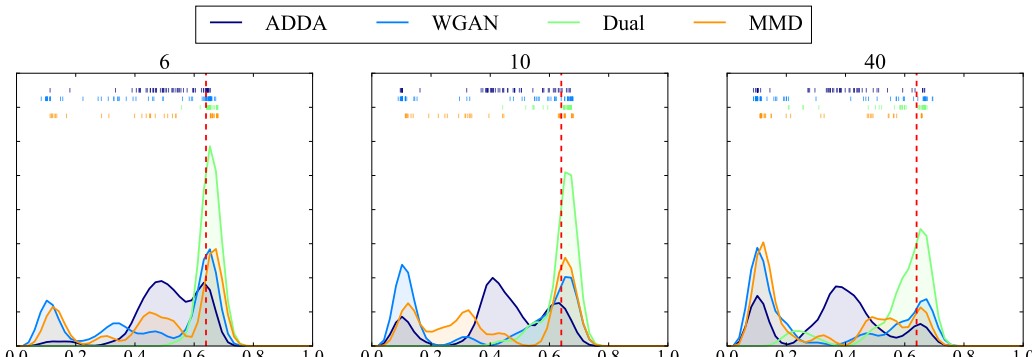

