# OpenReview forum: "Stable Distribution Alignment Using the Dual of the Adversarial Distance"
_ICLR.cc/2018/Conference — Invite to Workshop Track_

### Official Review · AnonReviewer2 · 2017-11-27
**A very specific and restricted fix to GANs**

**Rating:** 5
**Confidence:** 4

**Review:**

The paper deals with “fixing GANs at the computational level”, in a similar sprit to f-GANs and WGANs. The fix is very specific and restricted. It relies on the logistic regression model as the discriminator, and the dual formulation of logistic regression by Jaakkola and Haussler.

Comments:
1) Experiments are performed by restricting alternatives to also use a linear classifier for the discriminator. It is mentioned that results are expected to be lower than those produced by methods with a multi-layer classifier as the discriminator (e.g. Shen et al., Wasserstein distance guided representation learning for domain adaptation, Ganin et al., Domain-adversarial training of neural networks?).
2) Considering this is an unsupervised domain adaption problem, how do you set the hyper-parameters lambda and the kernel width? The “reverse validation” method described in Ganin et al., Domain-adversarial training of neural networks, JMLR, 2016 might be helpful.

Minor comments: on the upper-bound of the distance, alpha_i instead of alpha^\top, and please label the axes in your figures.

---

> ### Author Response · Authors · 2017-12-20
>
> 1. “<..> Minor comments: on the upper-bound of the distance, alpha_i instead of alpha^\top, and please label the axes in your figures. “ - fixed (eq. 2-3, pp. 6, 8)
>
> 2. “<..> Experiments are performed by restricting alternatives to also use a linear classifier for the discriminator.” - all three reviewers mentioned this, so we considered writing a single reply and put it into the reply section at the top of the page.
>
> 3. “<..> Considering this is an unsupervised domain adaptation problem, how do you set the hyper-parameters lambda and the kernel width?” - we first chose them semi-manually by validating on target, and then tested performance of all models with all hyperparameters spanning a grid between maximal and minimal optimal values from the first step; exact values are given in the supplementary.

---

### Official Review · AnonReviewer1 · 2017-11-27
**An interesting dual formulation of an adversarial loss, but its applicability remains to be fully demonstrated**

**Rating:** 6
**Confidence:** 4

**Review:**

This paper studies a dual formulation of an adversarial loss based on an upper-bound of the logistic loss. This allows the authors to turn the standard min max problem of adversarial training into a single minimization problem, which is easier to solve. The method is demonstrated on a toy example and on the task of unsupervised domain adaptation.

Strengths:
- The derivation of the dual formulation is novel
- The dual formulation simplifies adversarial training
- The experiments show the better behavior of the method compared to adversarial training for domain adaptation

Weaknesses:
- It is unclear that this idea would generalize beyond a logistic regression classifier, which might limit its applicability in practice
- It would have been nice to see results on other tasks than domain adaptation, such as synthetic image generation, for which GANs are often used
- It would be interesting to see if the DA results with a kernel classifier are better (comparable to the state of the art)
- The mathematical derivations have some errors


Detailed comments:
- The upper bound used to derive the formulation applies to a logistic regression classifier. While effective, such a classifier might not be as powerful as multi-layer architectures that are used as discriminators. I would be interested to know if they authors see ways to generalize to better classifiers.

- The second weakness listed above might be related to the first one. Did the authors tried their approach to non-DA tasks, such as generating images, as often done with GANs? Showing such results would be more convincing. However, I wonder if the fact that the method has to rely on a simple classifier does not limit its ability to tackle other tasks.

- The DA results are shown with a linear classifier, for the comparison to the baselines to be fair, which I appreciate. However, to evaluate the effectiveness of the method, it would be interesting to also report results with a kernel-based classifier, so as to see how it compares to the state of the art.

- There are some errors and unclear things in the mathematical derivations:
* In the equation above Eq. 2, \alpha should in fact be \alpha_i, and it is not a vector (no need to transpose it)
* In Eq. 2, it should be \alpha_i \alpha_j instead of \alpha^T\alpha
* In Eq. 3, it is unclear to me where the constraint 0 \leq \alpha \leq 1 comes from. The origin of the last equality constraints on the sums of \alpha_A and \alpha_B is also unclear to me.
* In Eq. 3, it is also not clear to me why the third term has a different constant weight than the first two. This would have an impact on the relationship to the MMD

- The idea of sample reweighting within the MMD was in fact already used for DA, e.g., Huang et al., NIPS 2007, Gong et al., ICML 2013. What is done here is quite different, but I think it would be worth discussing these relationships in the paper.

- The paper is reasonably clear, but could be improved with some more details on the mathematical derivations (e.g., explaining where the constraints on \alpha come from), and on the experiments (it is not entirely clear how the distributions of accuracies were obtained).

---

> ### Author Response · Authors · 2017-12-20
>
> 1. “<..> It is unclear that this idea would generalize beyond a logistic regression classifier, which might limit its applicability in practice”
>
> All three reviewers mentioned this, please see the reply section at the top of the page.
>
> 2. “<..> there are some errors and unclear things in the mathematical derivations”
>
> Thank you for paying such close attention to our derivations and pointing to these issues. [0, 1]-constraints on \alpha values are required by the upper bound - otherwise it would not hold; equation 3 illustrates computing a quadratic form in block form by putting down terms that correspond to interactions between points from same and different domains separately; the interaction is symmetric, therefore there are two identical cross-domain terms that results in a multiplier of two in front of that term. We added these details (page 4).
>
> The constraint on alpha sums (sum over A = sum over B) results from optimality conditions on the bias term of the discriminator - and it indeed does not quite follow from our derivations in their current form, you are completely right. We also added that.
>
> 3. “<..> - The idea of sample reweighting within the MMD was in fact already used for DA, e.g., Huang et al., NIPS 2007, Gong et al., ICML 2013. What is done here is quite different, but I think it would be worth discussing these relationships in the paper.”
>
> Indeed, Gong et al. were optimizing a sample reweighted MMD but restricted weights of points from one of domains to always equal one (1.0). They had somewhat different reasoning, as they were choosing points for auxiliary tasks at multiple scales, therefore, no iterative reweighting was performed. But, yes, thank you for these references, they indeed used a very similar idea and we referenced this paper in the related work section.
>
> Unfortunately, we were unable to find a paper you referred as “Huang et al., NIPS 2007”, do you think you could remember its actual name? It would be very interesting to read if it was as closely related to our work as Gong et al.
>
> 4. “<..> it is not entirely clear how the distributions of accuracies were obtained”
>
> Could you please comment on what exactly was not entirely clear? We chose reasonable values of hyperparameters for each model by hand and them performed a grid search for values “in-between”. Plotted distributions present how accuracies changed for different runs. More details are given in the supplementary.

---

> > ### Comment · AnonReviewer1 · 2017-12-21
> > **No title**
> >
> > 3. Huang et al., NIPS 2007:
> > The title is Correcting Sample Selection Bias by Unlabeled Data
> >
> > 4. You explanation answers my question. I was not sure how DISTRIBUTIONS of accuracies were obtained, as opposed to just accuracies.

---

> > > ### Author Response · Authors · 2018-01-01
> > >
> > > Indeed, there seem to be a relationship between instance reweighting approach, theoretically justified and empirically shown to work in the presence of domain shift before, and iteratively reweighted discrimination from this paper, thank you for pointing us towards this! We addressed this in the related work section (page 2) in the last revision. We want to stress again, that the main proposition of the paper is not to "use an iteratively reweighted MMD" - this is only a special case of a more general idea that suggests replacing a "competing" objective with a cooperative "correspondence" objective - and success of instance reweighting of training samples seems to further confirm applicability of the suggested family of "correspondence" objectives. We discussed this in more details in the comment section above titled "To all reviewers".

---

> ### Author Response · Authors · 2018-01-05
>
> To show that our approach works with non-linear discriminators, we ran a series of experiments with kernel discriminators. Our dual approach achieved similar but more stable results to those achieved by Deep Adaptation Networks (DAN) [1] which uses kernelized MMD. On the SVHN->MNIST shift our dual method obtains top accuracy of 70%, on par with DAN results reported in [2,3].  We tried both single fixed, multiple fixed and multiple varying kernels updated via gradient descent with different learning rates and kernel bandwidth values. Similar more stable behavior was observed when training our dual analog of Domain Confusion [4] method, as reported in the paper. This further confirms our main hypothesis that our proposed cooperative formulation of the problem via optimal alignment lead to better stability of the resulting distribution alignment method wrt the variation of hyperparameters.
>
> [1] Mingsheng Long, Yue Cao, Jianmin Wang, and Michael Jordan. "Learning transferable features with deep adaptation networks." ICML 2015
> [2] Kuniaki Saito, Yoshitaka Ushiku, Tatsuya Harada “Asymmetric Tri-training for Unsupervised Domain Adaptation” CoRR 2017
> [3] Philip Haeusser, Thomas Frerix, Alexander Mordvintsev, Daniel Cremers “Associative Domain Adaptation” CoRR 2017
> [4] Eric Tzeng, Judy Hoffman, Trevor Darrell, Kate Saenko “Simultaneous Deep Transfer Across Domains and Tasks” ICCV 2015

---

### Official Review · AnonReviewer3 · 2017-12-01
**An interesting but limited dual approach to adversarial distance**

**Rating:** 6
**Confidence:** 3

**Review:**

This paper proposes to re-formulate the GAN saddle point objective (for a logistic regression discriminator) as a minimization problem by dualizing the maximum likelihood objective for regularized logistic regression (where the dual function can be obtained in closed form when the discriminator is linear). They motivate their approach by repeating the previously made claim that the naive gradient approach is non-convergent for generic saddle point problems (Figure 1); while a gradient approach often works well for a minimization formulation.

The dual problem of regularized logistic regression is an entropy-regularized concave quadratic objective problem where the Hessian is y_i y_j <x_i, x_j>, thus highlighting the pairwise similarities between the points x_i & x_j; here the labels represent whether the point x comes from the samples A from the target distribution or B from the proposal distribution. This paper then compare this objective with the MMD distance between the samples A & B. It points out that the adversarial logistic distance can be viewed as an iteratively reweighted empirical estimator of the MMD distance, an interesting analogy (but also showing the limited power of the adversarial logistic distance for getting good generating distributions, given e.g. that the MMD has been observed in the past to perform poorly for face generation [Dziugaite et al. UAI 2015]). From this analogy, one could expect the method to improves over MMD, but not necessarily significantly in comparison to an approach which would use more powerful discriminators.

This paper then investigates the behavior of this adversarial logistic distance in the context of aligning distributions for domain adaptation, with experiments on a visual adaptation task. They observe better performance for their approach in comparison to ADDA, improved WGAN and MMD, when restricting the discriminators to be a linear classifier.

== Evaluation

I found this paper quite clear to read and enjoyed reading it. The observations are interesting, despite being on the toyish side. I am not an expert on GANs for domain adaptation, and thus I can not judge of the quality of the experimental comparison for this application, but it seemed reasonable, apart for the restriction to the linear discriminators (which is required by the framework of this paper).

One concern about the paper (but this is an unfortunate common feature of most GAN papers) is that it ignores the vast knowledge on saddle point optimization coming from the optimization community. The instability of a gradient method on non-strongly convex-concave saddle point problems (like the bilinear form of Figure 1) is a well-known property, and many alternative *stable* gradient based algorithms have been proposed to solve saddle point problems which do not require transforming them to a minimization problem as suggested in this paper. Moreover, the transformation to the minimization form crucially required the closed form computation of the dual function (with w* just defined above equation (2)), and this is limited to linear discriminators,  thus ruling out the use of the proposed approach to more powerful discriminators like deep neural nets. Thus the significance appears a bit limited to me.

== Other comments

1) Note that d(A, B'_theta) is *equal* to min_alpha max_w  (...)  above equation (2) (it is not just an upper bound). This is a standard result coming from the fact that the Fenchel dual problem to regularized maximum likelihood is the maximum entropy problem with a quadratic objective as (2).  See e.g. Section 2.2 of [Collins et al. JMLR 2008] (this is for the more general multiclass logistic regression problem, but (2) is just the binary special case of equation (4) in the [Collins ... ] reference). And note that the "w(u)" defined in this reference is the lambda*w*(alpha) optimal relationship defined in this paper (but without the 1/lambda factor because of just slightly different writing; the point though is that strong duality holds there and thus one really has equality).


[Collins et al. JMLR 2008] Michael Collins, Amir Globerson, Terry Koo , Xavier Carreras, Peter L. Bartlett, Exponentiated Gradient Algorithms for Conditional Random Fields and Max-Margin Markov Networks, , JMLR 2008.

 [Dziugaite et al. UAI 2015] Gintare Karolina Dziugaite, Daniel M. Roy, and Zoubin Ghahramani. Training generative neural networks via maximum mean discrepancy optimization. In UAI, 2015

---

> ### Author Response · Authors · 2017-12-20
> **To all reviewers:**
>
>  “<..> limited to linear discriminators,  thus ruling out the use of the proposed approach to more powerful discriminators like deep neural nets”
>  “<..> It is unclear that this idea would generalize beyond a logistic regression classifier, which might limit its applicability in practice”
> “<..> It is mentioned that results are expected to be lower than those produced by methods with a multi-layer classifier as the discriminator”
>
> The reviewers are correct in that our proposed dual formulation in its current form only applies to the logistic family of classifiers, however, this is not limited to linear classifiers, because we can use kernels to obtain a nonlinear classifier. For example, MMD-based methods also use kernels and obtain state of the art results for domain adaptation.
>
> While we agree that the exact dual exists only in the logistic discriminator case, when one can use the duality and solve the inner problem in closed form, we want to stress that our paper presents a more general framework for alignment that can be extended to other classes of functions. More specifically, one can rewrite the quadratic form in kernel logistic regression [eq. 3, page 4] as a Frobenius inner product of a kernel matrix Q with a symmetric rank 1 alignment matrix S (outer product of alpha with itself). The kernel matrix specifies distances between points and S chooses pairs that minimize the total distance. This way the problem reduces to “maximizing the maximum agreement between the alignment and similarity matrices” that in turn might be seen as replacing “adversity” in the original problem with “cooperation” in the dual maximization problem. In our paper, S is a rank 1 matrix, but we could choose different alignment matrix parameterizations that would correspond to having a neural network discriminator in the adversarial problem. It is not exactly dual to minimization of any existing adversarial distances, but exploits same underlying principle of “iteratively-reweighted alignment matrix fitting” discussed in this paper. We assert that in order to understand the basic properties of the resulting formulation, an in-depth discussion of the well-studied logistic case is no less important than the discussion involving complicated deep models, which deserves a paper of its own. This paper proposes a more stable “cooperative” problem reformulation rather than a new adversarial objective as many recent papers do.
>
> We added a section discussing relations to existing methods from this “cooperative point of view” and exciting future possibilities. (page 8)

---

> ### Author Response · Authors · 2017-12-20
>
> 1. “<..> ignores the vast knowledge on saddle point optimization coming from the optimization community”
>
> We want to thank you for pointing this out. We are concerned about the lack of focus on actual optimization methods in the context of GANs too. We added a paragraph on Mirror Descent and Fictitious Play (page 3). If you happen to have any other suggestions on methods to discuss in this context, please let us know.
>
> 2. “<..> the point though is that strong duality holds there and thus one really has equality”
>
> Completely correct, thank you. We put that inequality to emphasise the log-sigmoid upper bound, but considering that it is also tight for optimal choice of alpha anyway, this indeed must be very confusing, we changed that (eq. 2-3, p. 4).

---

### Decision · Program_Chairs · 2018-01-29
**ICLR 2018 Conference Acceptance Decision**

**Decision:**

Invite to Workshop Track

**Comment:**

All the reviewers noted that the dual formulation, as presented, only applies to the logistic family of classifiers. The kernelization is of course something that *can* be done, as argued by the authors, but is not in fact approached in the submission, only in the rebuttal. The toy-ish nature of the problems tackled in the submission limits the value of the presentation.

If the authors incorporate their domain adaptation results (SVHN-->MNIST and others) using the kernelized approach and do the stability analysis for those cases, and obtain reasonable results on domain adaptation benchmarks (70% on SVHN-->MNIST is for instance on the low side compared to the pixel-transfer-based GAN approaches out there!) then I think it'd be a great paper.

As such, I can only recommend it as an invitation to the workshop track, as the dual formulation is interesting and potentially useful.